# Intraspecific variation in immune gene expression and heritable symbiont density

**Holly L. Nichols**[1], **Elliott B. Goldstein**[1], **Omid Saleh Ziabari**[2], **Benjamin J. Parker**[1]*

1 Department of Microbiology, University of Tennessee, Knoxville, Tennessee, United States of America,
2 Department of Biology, University of Rochester, Rochester, New York, United States of America

* bjp@utk.edu

**Data Availability Statement:** All experimental data is included in a supplementary file. Sequence data has been uploaded to the NCBI Sequence Read Archive (SRA) with accession numbers PRJNA684046 and PRJNA684311.

## Abstract

Host genetic variation plays an important role in the structure and function of heritable microbial communities. Recent studies have shown that insects use immune mechanisms to regulate heritable symbionts. Here we test the hypothesis that variation in symbiont density among hosts is linked to intraspecific differences in the immune response to harboring symbionts. We show that pea aphids (*Acyrthosiphon pisum*) harboring the bacterial endosymbiont *Regiella insecticola* (but not all other species of symbionts) downregulate expression of key immune genes. We then functionally link immune expression with symbiont density using RNAi. The pea aphid species complex is comprised of multiple reproductively-isolated host plant-adapted populations. These 'biotypes' have distinct patterns of symbiont infections: for example, aphids from the *Trifolium* biotype are strongly associated with *Regiella*. Using RNAseq, we compare patterns of gene expression in response to *Regiella* in aphid genotypes from multiple biotypes, and we show that *Trifolium* aphids experience no downregulation of immune gene expression while hosting *Regiella* and harbor symbionts at lower densities. Using F1 hybrids between two biotypes, we find that symbiont density and immune gene expression are both intermediate in hybrids. We propose that in this system, *Regiella* symbionts are suppressing aphid immune mechanisms to increase their density, but that some hosts have adapted to prevent immune suppression in order to control symbiont numbers. This work therefore suggests that antagonistic coevolution can play a role in host-microbe interactions even when symbionts are transmitted vertically and provide a clear benefit to their hosts. The specific immune mechanisms that we find are downregulated in the presence of *Regiella* have been previously shown to combat pathogens in aphids, and thus this work also highlights the immune system's complex dual role in interacting with both beneficial and harmful microbes.

## Author summary

Insects frequently form beneficial partnerships with heritable microbes that are passed from mothers to offspring. Natural populations exhibit a great deal of variation in the frequency of heritable microbes and in the within-host density of these infections. Uncovering the mechanisms underlying variation in host-microbe interactions is key to

**Funding:** This work was funded by startup funds from the University of Tennessee to BJP. The funder had no role in study design, data collection and analysis, decision to publish, or preparation of the manuscript.

**Competing interests:** The authors have declared that no competing interests exist.

understanding how they evolve. We study a model host-microbe interaction: the pea aphid and a heritable bacterium that makes aphids resistant to fungal pathogens. We show that aphids harboring bacteria show sharply reduced expression of innate immune system genes, and that this leads to increased densities of symbionts. We further show that populations of aphids that live on different species of plants vary in differential immune gene expression and in the density of their symbiont infections. This study contributes to our mechanistic understanding of an important model of host-microbe symbiosis and suggests that hosts and heritable microbes are evolving antagonistically. This work also sheds light on how invertebrate immune systems evolve to manage the complex task of combatting harmful pathogens while accommodating potentially beneficial microbes.

## Introduction

Most insects harbor heritable microbes that have important effects on host fitness [1–3]. A key aspect of these symbioses is variation. Across species, host taxonomy has been shown to play a role in structuring heritable microbial communities [4–6]. Within species, microbes referred to as facultative symbionts are not found in all individuals, and symbiont frequencies are subject to selection on the relative costs and benefits of harboring microbes [7,8]. In addition to microbiome composition, hosts vary in other aspects of symbioses like the density of microbial infections [9,10]. For example, two closely-related species of *Nasonia* wasps vary in the density at which they harbor *Wolbachia* bacteria, and this variation is due to a single gene that somehow suppresses maternal transmission of bacteria [11]. Except for a few examples, little is known about the mechanisms that underlie variation in heritable symbioses or the evolutionary genetics of these interactions [12].

Invertebrate immune systems have been shown to play a direct role in mediating interactions with heritable microbes. In grain weevils, for example, an antimicrobial peptide acts to confine mutualistic symbionts to specialized cells called bacteriocytes [13], and silencing expression of immune pathways allows symbionts to escape host cells [14]. Other studies have found more complex interactions between pathogens, the immune system, and vertically-transmitted symbionts. In *Drosophila melanogaster*, for example, activation of the Toll and IMD pathways results in an increase in density of *Spiroplasma* symbionts [15] (and see similar examples in mosquitos [16] and tsetse flies [17]), suggesting in some systems the immune system can promote beneficial symbionts by inhibiting other microbes.

Immune genes are among the fastest evolving in eukaryotic genomes [18–20], and natural populations harbor extensive genetic variation in immune mechanisms [21]. Given the importance of the immune system in regulating insect symbioses, it seems likely that variation in immune mechanisms contributes to variation in symbiont density among hosts. This hypothesis is complicated, however, because our models for the maintenance of genetic variation in immune systems are based on antagonistic coevolution between hosts and pathogenic microbes [22,23]. Heritable symbiont infections are thought to spread through host populations because the fitness interests of hosts and microbes are generally aligned, and many symbionts have been shown to benefit their hosts for example by providing protection from pathogens (reviewed in [24]). But symbionts can impose costs on their hosts (e.g. [25]), and selection may favor the loss of symbionts in certain contexts [26]. In addition, within-host selection might lead to a separation of the fitness interests of hosts and microbes. For example, a mutation in a symbiont genome that increases symbiont density might increase the

likelihood of symbiont transmission but come at the expense of host fitness. Hosts, in turn, could evolve greater control over symbiont numbers in an ongoing arms-race for control over a symbiosis. It is unclear, however, whether the 'arms-race' dynamics underlying host-pathogen coevolution also govern the evolutionary interactions between immune systems and beneficial microbes.

The pea aphid (*Acyrthosiphon pisum*) is an important insect-symbiont model system [27]. The pea aphid species complex is composed of multiple reproductively-isolated populations adapted to live on different host plants within the family Fabaceae. These biotypes are genetically differentiated and are estimated to have radiated onto different host plants ~500,000 years ago [28] (but see [29]). In addition to obligate intracellular bacteria called *Buchnera aphidicola*, aphids can harbor several species of facultative symbionts. Multiple studies have found that facultative symbionts are non-randomly distributed across aphid biotypes [30–33]. For example, *Regiella insecticola* (which confers protection against fungal pathogens to its host [34–36]) is strongly associated with aphids from the *Trifolium* spp. (clover) biotype across continents. A number of studies have explored whether the strong association between *Trifolium* biotype aphids and *Regiella* is due to improved host plant use with mixed results [37–40]. Alternatively, this association could be due to the risk of exposure to fungal pathogens (though see [41]), to historical contingency (though see [42]), or to host and/or symbiont genetic mechanisms. Beyond the species level, *Regiella* within pea aphids form two main phylogenetic clades, and biotype has been shown to be a significant factor underlying the distribution of *Regiella* strains among pea aphids [43]. Specifically, the strong association between *Regiella* and aphids from the *Trifolium* biotype is mainly driven by *Regiella* from one specific clade (termed 'Clade 2') [43]. This system therefore provides a useful natural laboratory to study host-microbe adaptation across multiple environments within a single host species.

In this study, we show that immune mechanisms play a role in intraspecific variation in the density of a heritable bacterial symbiont. We first demonstrate that pea aphids that harbor *Regiella* (but not all other species of symbionts) sharply downregulate key innate immune genes, and that experimental suppression of the immune gene phenoloxidase via RNAi increases symbiont density. We then measure gene expression across aphids from multiple biotypes and find that aphids from *Trifolium* do not experience immune gene downregulation and harbor symbionts at relatively low density. Finally, by performing an F1 cross between genotypes from two biotypes we find that hybrid aphids show intermediate symbiont densities and immune gene downregulation, shedding light on the role of host genetic variation and the genomic architecture of this variation. We discuss these findings in light of the biology of this system and suggest that antagonistic coevolution between 'beneficial' microbes and their hosts can shape host-symbiont associations.

## Results

### Hosting some symbiont species leads to decreased host immune gene expression

Aphid lines reproduce parthenogenetically under summer conditions, and facultative bacteria can be introduced into or removed from host lines. We established lines that have the same aphid host genotype (LSR1, collected from *Medicago sativa* [44]) with two different strains of *Regiella*: one from each of the two main phylogenetic clades of *Regiella* found in natural populations of pea aphids [43]. *Regiella* strain .LSR was originally collected in Ithaca, NY, USA in 1998 with the LSR1 aphid genotype and is a representative of *Regiella* Clade 1. *Regiella* strain .313 was collected in Gloucestershire, UK, in 2007 from an aphid from the *Trifolium* biotype (genotype 313) and is from Clade 2 [45]. We maintained aphids in the lab for four generations

after symbiont establishment and then measured *Regiella* densities using quantitative PCR (qPCR). We have found previously that Clade 2 *Regiella* establish at higher densities in hosts than do Clade 1 strains, independent of host genotype [46]. Consistent with this previous work, *Regiella* strain .313 established in aphid genotype LSR1 at a significantly higher density (3.5X) than *Regiella* strain .LSR (t = 5.1, p = 0.006, Fig 1A).

We then used RNAseq to measure how harboring *Regiella* influences aphid gene expression by comparing these lines with symbiont-free aphids that were sham-injected. We sequenced cDNA made from mRNA for 4 biological replicates of each line (where a biological replicate represents an independent aphid line injected with symbionts or sham-injected). Overall, harboring *Regiella* strain .LSR did not significantly alter expression of any genes in the aphid

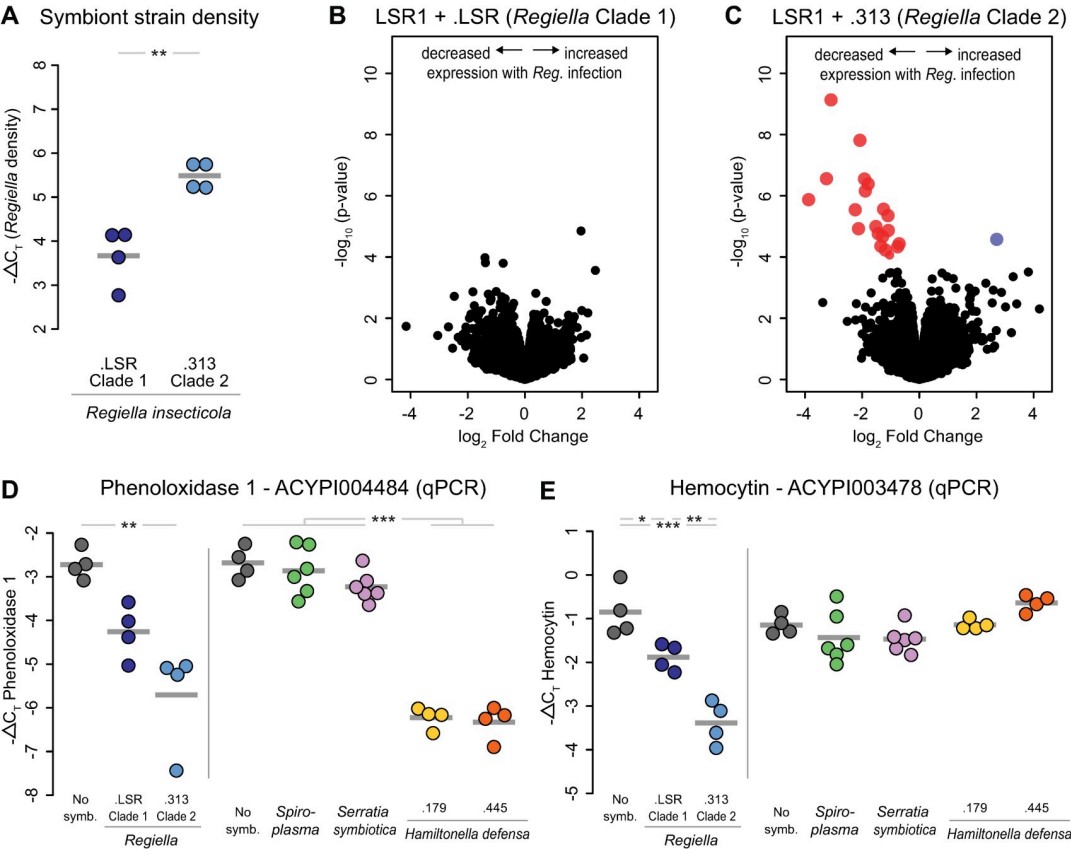

**Fig 1. Effects of hosting facultative symbionts on aphid gene expression. A:** Density of lines harboring a Clade 1 (.LSR) and a Clade 2 (.313) strain of *Regiella*. The y-axis shows the -$\Delta C_T$ values which can be interpreted on a $\log_2$ scale. The average -$\Delta C_T$ value for each *Regiella* strain is shown with a grey bar. Statistical significance (t-test) is shown along the top of the figure. **B&C:** Volcano plots of RNAseq data comparing expression of each expressed gene in the aphid genome, represented by a point in each figure, between aphids with and without *Regiella*. B and C show this analysis for aphids with a *Regiella* strain from Clade 1 and Clade 2, respectively. The x-axes show the $\log_2$ fold change for each gene, and the y-axis shows the significance of expression. Blue and red dots represent genes that were significantly up- or down-regulated, respectively, at an FDR < 0.05. **D:** qPCR analysis of gene expression of phenoloxidase 1 (*PO1*; ACYPI004484) in response to different species of facultative symbionts. The grey dots represent aphids without facultative symbionts, and the colored dots show those with symbiont infections 4 generations after symbiont establishment. The average -$\Delta C_T$ value for each group is shown with a grey bar. The different symbiont species and strains are shown along the bottom of the figure. The y-axis shows the -$\Delta C_T$ values of expression, which can be interpreted on a $\log_2$ scale. Statistical significance among species or strains (ANOVA & Tukey's HSD post-hoc analysis) is shown along the top of the figure; the two experiments, which are separated by a vertical line, were analyzed separately. **E:** Same as panel D but for the gene hemocytin (ACYPI003478).

genome (FDR < 0.05; Fig 1B), while strain .313 led to significantly decreased expression of 19 genes and upregulation of 1 gene (FDR < 0.05; Fig 1C).

The 20 genes with altered expression included key innate immune system genes (S1 Table). In particular, the two copies of phenoloxidase in the pea aphid genome (referred to here as *PO1* and *PO2*) were downregulated in the presence of *Regiella*. Also downregulated was a gene called hemocytin, which encodes a protein released by immune cells that plays a role in immune cell aggregation [47,48]. Other differentially expressed genes included a toll-like receptor and a putative lipopolysaccharide recognition protein (S1 Table).

We used qPCR to directly compare expression of two immune genes between lines harboring the two *Regiella* strains and to confirm our RNAseq results. *PO1* was significantly downregulated in lines harboring *Regiella* strain .313 (Fig 1D and 1E, left panels). Hemocytin was significantly downregulated in aphids harboring either symbiont, and the magnitude of this change was significantly stronger for aphids harboring strain .313 than those with strain .LSR (Fig 1D and 1E, left panels).

Next, we established aphid lines that harbored one of several additional species of aphid facultative symbionts as above, and we looked for changes in *PO1* and hemocytin expression. *Spiroplasma* and *Serratia symbiotica* did not alter expression of either gene, but two strains of *Hamiltonella defensa* significantly downregulated *PO1* expression but not hemocytin (post-hoc tests, S2 Table and Fig 1D and 1E, right panels). Like *Regiella*, the specific strain of *Spiroplasma* (.161) we used was found in previous work to protect against fungal pathogens [49], but *Serratia* and *Hamiltonella* have been shown not to influence fungal resistance [36,50,51]. These results therefore suggest that the changes we identify in immune gene expression do not reflect the mechanism by which *Regiella* confers protection to aphids against fungal pathogens, which is currently unknown. Aphid symbionts in the family Enterobacteriaceae (*Regiella*, *Hamiltonella*, and *Serratia*) live both in hemolymph and inside of the insect cells (reviewed in [52]). In contrast, *Spiroplasma* is mainly extracellular in most insects, and has not been identified inside of aphid cells [53]. Qualitatively, then, our results do not suggest that differences in immune gene expression among facultative symbiont species are due to localization in different tissues.

## Immune gene expression influences symbiont density during development

We studied the function of immune gene expression on *Regiella* densities using RNA interference (RNAi) [54]. We knocked-down expression of *PO1* early in development by injecting ~100ng of dsRNA in salt buffer into 1-day-old aphids, and we measured the effects on symbiont density. We used dsRNA for *lacZ* as a control, which is designed to control for the effects of injection and exposure to buffer and dsRNA, which may have an influence on gene expression. Aphids harbored either *Regiella* strain .LSR or .313. We then sampled aphids at two timepoints: at 72hrs after injection and after aphids had become adults (8 days after injection).

Injection with *PO1* dsRNA reduced *PO1* expression, on average, by ~60% at 72hrs (2-way ANOVA; Treatment: F = 9.8, p = 0.009; Fig 2A). At this early timepoint, we found no significant difference between aphids harboring *Regiella* strain .LSR vs strain .313 in *PO1* expression (Strain: F = 0.10, p = 0.75; Fig 2A). By the time aphids became adults (8 days after injection), *PO1* expression in *PO1* dsRNA injected aphids was still reduced by ~60% compared with controls (Treatment: F = 3.6, p = 0.02; Fig 2A). By this later timepoint, aphids harboring the two symbiont strains had diverged in expression as found above (Strain: F = 11.5, p = 0.004; Fig 2A). Note that we dissected out and removed developing embryos only from the adult samples before nucleic acid extraction, so we do not directly compare expression in the 72hrs vs. adult samples, but qualitatively *PO1* expression increased during development (Fig 2A).

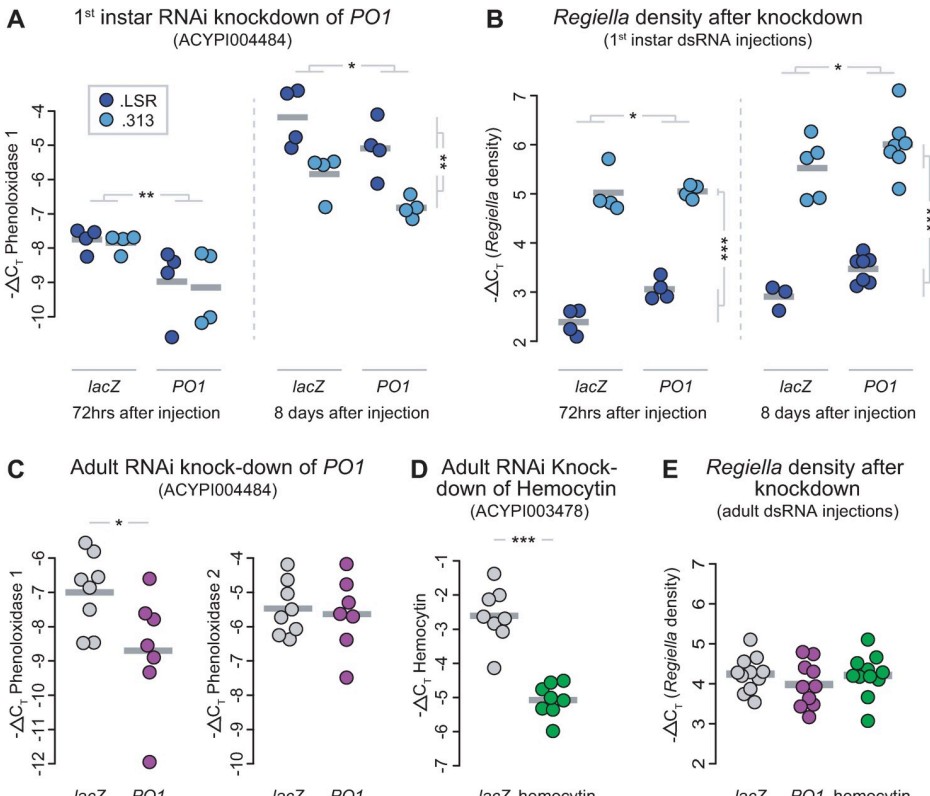

**Fig 2. RNAi knockdowns and *Regiella* density. A:** Validation of the RNAi knockdown for *PO1* in 1st instar aphids. The y-axis shows $-\Delta C_T$ values of expression, which can be interpreted on a log₂ scale. Collection time-points (72 hours or 8 days) and treatment are shown along the bottom of the figure; the two *Regiella* strains are represented by different colors as shown in the key. Grey bars show the mean of each treatment group, with biological replicates shown as points. Statistical significance (2-way ANOVA & Tukey's HSD post-hoc analysis) among treatment groups (*lacZ* vs. *PO1*) is shown along the top of each plot, with an *, **, or *** indicating altered gene expression at p < 0.05, p < 0.01, or p < 0.001, respectively. Significance among *Regiella* genotypes (.LSR vs .313) is shown along the right side of each plot. **B:** *Regiella* density after knockdown of aphids injected as 1st instars. The y-axis shows $-\Delta C_T$ values of symbiont density, which can be interpreted on a log₂ scale. Time-points, treatment, *Regiella* strain, and statistical significances (2-way ANOVA & Tukey's HSD post-hoc analysis) are indicated as in A. **C:** Validation of the RNAi knockdown for *PO1* in adult aphids. Y-axes are as above, treatment is shown along the bottom of the figures, with lacZ and *PO1* dsRNA-injected aphids represented by grey and purple dots, respectively. Grey gars shown mean expression for a treatment. The left panel shows expression of *PO1*, and the right panel shows expression of the other copy of phenoloxidase in the aphid genotype (*PO2*). Statistical significances (t-tests) are shown as above. **D:** Knock-down validation of hemocytin. **E:** *Regiella* density after knockdown of aphids injected as adults. Grey, purple, and green points represent different treatments (*lacZ*, *PO1*, and hemocytin dsRNA injections, respectively). The y-axis shows *Regiella* density measured by $-\Delta C_T$ values as above.

*PO1* knockdown led to a 59% and 2% increase in *Regiella* strain .LSR and .313 density at 72hrs, respectively (2-way ANOVA; Treatment: F = 4.7, p = 0.05; Fig 2B). *Regiella* density in aphids harboring strain .LSR vs .313 also differed significantly (Strain: F = 245, p < 0.0001; Fig 2B), suggesting that strain-level differences in symbiont density are present even at this early developmental timepoint. The increase in *Regiella* density due to *PO1* knock-down persisted to adulthood (Treatment: F = 5.7, p = 0.03, Fig 2B), with *PO1* dsRNA injection increasing *Regiella* density by 48% and 40% in aphids harboring *Regiella* strains .LSR and .313, respectively. As we found above, the density of strain .313 was higher than strain .LSR in adult aphids (Strain: F = 145, p < 0.0001; Fig 2B). Together, these results show that knockdown of *PO1* increases *Regiella* density over development.

## Symbiont density is not impacted by immune gene knock-down later in development

We performed a similar experiment studying the effect of RNAi on symbiont density, but injected dsRNA into adult aphids rather than 1st instars. We injected ~1μg of dsRNA synthesized from *PO1* or hemocytin into adult (9 day old) aphid genotype LSR1 aphids infected with *Regiella* strain .LSR. We measured gene expression and *Regiella* density at 72hrs after injection. This led to a ~69% and ~82% reduction in expression of *PO1* and hemocytin, respectively (t-tests; *PO1*: t = -2.3, p = 0.05; hemocytin: t = -7.3, p < 0.0001; Fig 1C and 1D). We note that injection with dsRNA from *PO1* had no effect on expression of the other copy of phenoloxidase in the aphid genome (*PO2*: t = -0.91, p = 0.38; Fig 1C), demonstrating that our phenoloxidase RNAi assay is specific to *PO1* as designed. Knockdowns had no effect on *Regiella* density in aphids injected as adults (ANOVA; Treatment: F = 0.80, p = 0.46; Fig 1E).

## Immune gene downregulation differs across aphid biotypes

We repeated the RNAseq experiment to study the effects of *Regiella* on gene expression across multiple aphid biotypes. We used a genotype from the *Lotus corniculatus* biotype (663, collected in Oxfordshire, UK in 2014 with no original facultative symbionts), a genotype from *Ononis spinosa* (C133, collected in Berkshire, UK in 2003 that originally harbored *Hamiltonella*), and a genotype from *Trifolium* spp. (C317, collected from *Trifolium pratense* in Glouchestershire, UK in 2003 that originally harbored a Clade 2 *Regiella*). The genetic distance among these biotypes is variable, with *Trifolium* and *Medicago sativa* (LSR1) possibly the most closely related and *Ononis* the most distant; unlike some biotypes (e.g. *Lathyrus pratensis*), those included here are not thought to represent incipient species [29]). For each aphid genotype, we compared replicate lines that had each been infected with an independent *Regiella* Clade 2 (.313) infection or had been sham injected as above after 4 generations (after verifying that the symbiont infection had not been lost using PCR).

In the *Lotus corniculatus* genotype, harboring *Regiella* strain .313 had a significant effect on the expression of aphid genes (7 genes differentially expressed at an FDR < 0.05; Fig 3A). Of these 7 genes, five were also downregulated in the experiment described above using genotype LSR1, including *PO1* and hemocytin (Fig 3D). There is therefore some degree of conservation in the response to *Regiella* across genetically distinct aphid lines (Fig 3D). In contrast, zero genes were differentially expressed in response to *Regiella* in the *Ononis* biotype line at an FDR of < 0.05 (Fig 3B). Similarly, zero genes differed in expression in response to *Regiella* in the *Trifolium* line (Fig 3C).

## F1 hybrids have an intermediate symbiont density and level of immune expression

We performed an F1 cross (Fig 4A) between two of the biotype lines in order to better understand the role of host genetic variation in *Regiella* density and immune gene expression. We crossed the *Lotus* (663) and *Trifolium* (C317) lines [55], and generated multiple replicate infections with *Regiella* strain .313 in each line as above. After four generations, we measured *Regiella* density using qPCR. *Regiella* density differed between the parental lines (post-hoc tests S3 Table and Fig 4B), with the *Lotus* line harboring a significantly higher density of *Regiella* strain .313 (5.2X higher) than the *Trifolium* line. Further, the F1 lines harbored *Regiella* at densities intermediate to the parental lines (post-hoc tests S3 Table and Fig 4B).

We then sampled aphids from this same generation to compare changes in immune gene expression due to *Regiella* in parental and F1 lines. We selected two F1 lines for this assay with

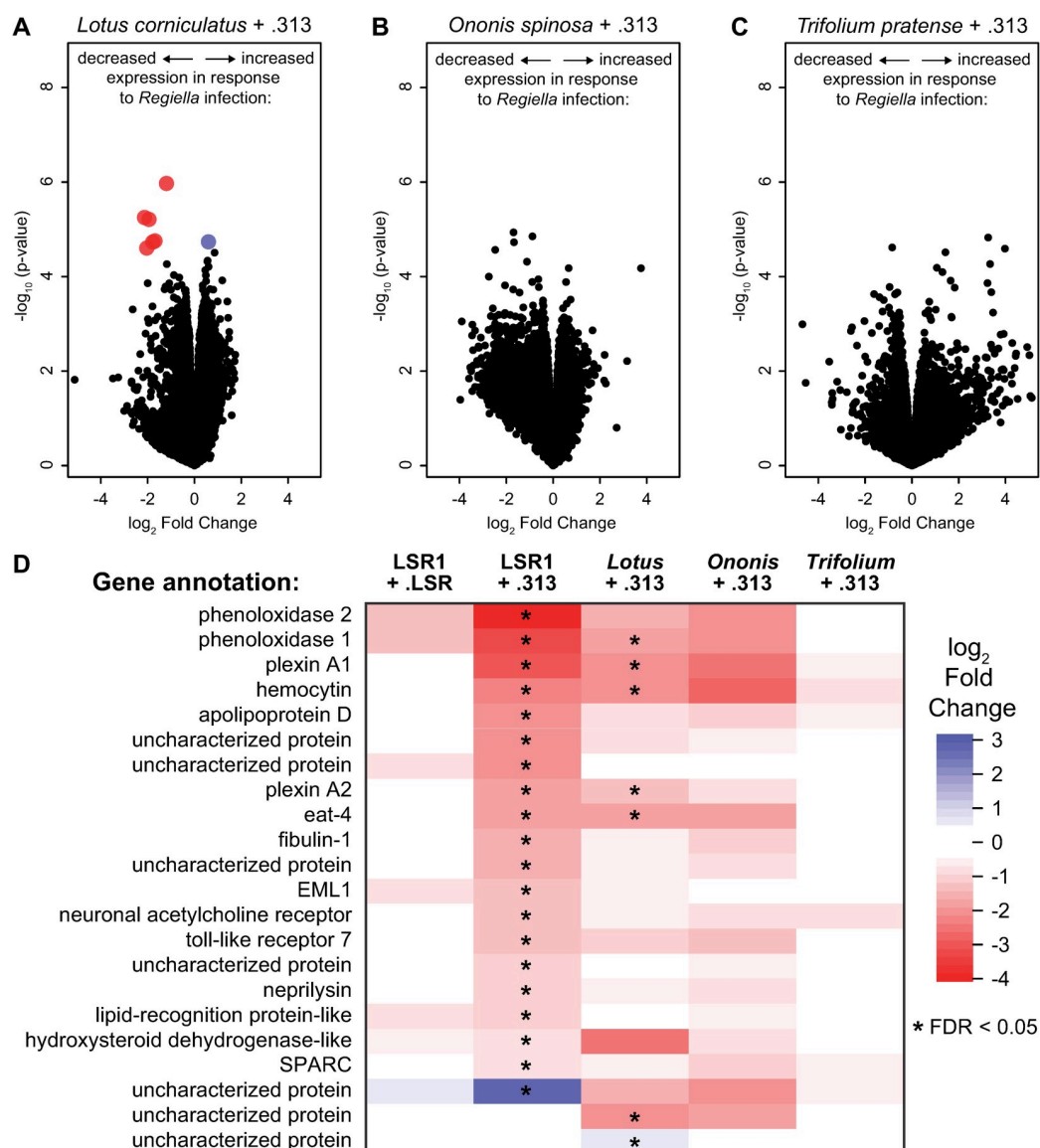

**Fig 3. Gene expression across aphid biotypes. A-C:** Volcano plots of expression data comparing control vs. *Regiella*-infected aphids. Each expressed gene in the aphid genome is represented by a point. The x-axes show the $\log_2$ fold change of each gene, with points to the right side of each plot indicating increased expression in the presence of symbionts, and points to the left showing decreased expression. The y-axes show the $-\log_{10}$ of the p-values indicating statistical significance of each gene's expression change. Colored points are those where the expression change was found to be statistically significant at FDR < 0.05. Panels A, B, and C show plots for the *Lotus*, *Ononis*, and *Trifolium* genotypes respectively, as shown along the top of the figures. **D:** A heat-map comparing gene expression in response to *Regiella* strain .313 infection across host genotypes. The 22 differentially expressed genes identified in the LSR1 transcriptome, above, are listed to the left of the figure. Colors represent the $\log_2$ fold change of these genes in response to *Regiella* as indicated in the key to the right of the figure (with red panels representing a decrease in expression, and blue indicating an increase in expression). The five transcriptomes generated in this study are shown in each column, as indicated at the top of the figure. Statistical significance of each gene is indicated by an asterisk at an FDR < 0.05.

each aphid genotype serving as the maternal line. We used qPCR to measure expression of both copies of phenoloxidase, hemocytin, and also nitric oxide synthase (*NOS*; an important innate immune mechanism that was not significantly differentially expressed in any of our RNAseq studies). Confirming our RNAseq findings, harboring *Regiella* led to a decrease in expression of *PO1*, *PO2*, and hemocytin in the *Lotus* genotype, but *Regiella* did not affect gene

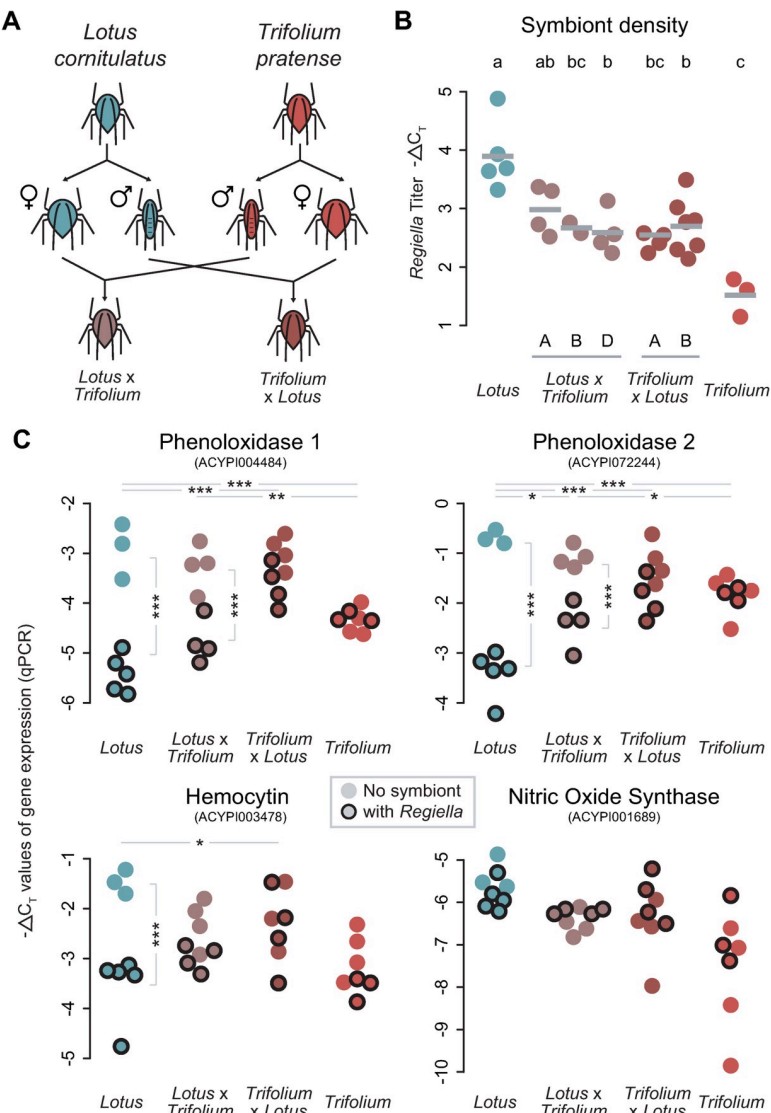

**Fig 4. *Regiella* density and immune gene expression in F1 hybrid lines. A:** Diagram of the crossing scheme. **B:** *Regiella* density in parental and F1 lines. The y-axis shows the -$\Delta C_T$ values reflecting *Regiella* density, which can be interpreted on a log$_2$ scale. The genotypes are shown along the bottom of the figure; the letters represent different replicate lines of each direction of the F1 cross. Each biological replicate, representing an independently injected aphid + *Regiella* line, is shown by a colored point, with the means for each genotype shown with grey bars. Significance groups (ANOVA & Tukey's HSD post-hoc analysis) are shown along the top of the figure at p < 0.05. **C:** Immune gene expression in lines with and without *Regiella*. The y-axis of each plot shows -$\Delta C_T$ values (qPCR output) of gene expression. Aphid genotype is shown along the bottom of each figure. Each biological replicate (an independently injected line) is shown by a colored point, with lines harboring *Regiella* indicated by a dark black outline as indicated in the legend. Statistical significance (2-way ANOVA & post-hoc analysis) is shown with light grey lines: at the top of the figures, the interaction term between *Regiella* presence/absence and host genotype is indicated with an *, **, or *** indicating that two genotypes differ in the extent to which *Regiella* altered gene expression at p < 0.05, p < 0.01, or p < 0.001, respectively. Within genotypes, whether *Regiella* significantly altered gene expression is shown with a vertical grey bracket.

expression in the *Trifolium* genotype (post-hoc tests S4 Table and Fig 4C). Further, the F1 lines showed significant differences in gene expression that were intermediate to the two parental lines: three immune genes were downregulated in response to *Regiella*, but to a significantly lesser extent than in the *Lotus* genotype (post-hoc tests S4 Table and Fig 4C).

## Discussion

We show that some aphids harboring the facultative bacterial symbiont *Regiella insecticola* experience reduced expression of key immune genes, and we link decreased immune gene expression with increases in *Regiella* density. We further find that this mechanism is influenced by host genetic factors: some genotypes harbor *Regiella* at lower densities and do not experience altered gene expression when infected, where other aphid genotypes experience reduced immune gene expression and have higher-density *Regiella* infections. This study shows that intraspecific variation in the immune system affects a heritable symbiosis.

Clade 2 *Regiella* establish in aphids at higher densities than Clade 1 strains [46]. We found no significantly differentially expressed genes in the transcriptome of aphids harboring a Clade 1 (.LSR) *Regiella* strain. Using qPCR we confirmed that the immune downregulation we uncover occurs more strongly in aphids harboring the Clade 2 *Regiella* than the Clade 1 strain, potentially contributing to differences in density among *Regiella* strains. An important question, then, is whether symbionts are suppressing host immune mechanisms in order to reach higher densities in hosts, or if hosts are modifying immune mechanisms in order to accommodate symbionts. Fitness costs to aphids of harboring symbionts (including *Regiella*) have been measured in the laboratory and field [35,56], and we have found previously that higher density Clade 2 *Regiella* strains impose stronger survival costs on hosts than the lower density Clade 1 strains [46]. In addition, the two *Regiella* clades confer protection against specific genotypes of the fungal pathogen *Pandora neoaphidis* [45], and therefore *Regiella* density will be positively correlated with symbiont-mediated protection for some fungal genotypes and negatively correlated for others. Together these results suggest that immune downregulation is not an adaptation on the part of the host in order to accommodate symbionts, but instead some *Regiella* strains suppress immune mechanisms in order to establish at higher densities in hosts. Establishing at a higher density could benefit *Regiella* through improved competitive outcomes with other strains and species of symbionts, or through increased horizontal transmission which occurs on evolutionary [43] and even ecological [57] timescales.

Aphid biotypes harbor facultative symbionts at different frequencies that are to some degree conserved across continents [30–33], and decades of research have gone into explaining these patterns in order to better understand the ecological and evolutionary forces shaping beneficial host-microbe interactions. One particularly clear association is between aphids from the *Trifolium* biotype and *Regiella*—and specifically *Regiella* from Clade 2 [43]. Studies attempting to explain this pattern have considered factors like the potential effects of *Regiella* on host plant use [37–40] and pressure from fungal pathogens on different host plants [41]. We suggest that host genetic effects represent an additional factor shaping the aphid-symbiont frequencies in natural populations. Whether a cause or consequence of the strong association between *Regiella* and *Trifolium*, it seems likely that that the immune systems of *Trifolium* aphids are better adapted to harboring this symbiont than biotypes that are not naturally associated with Clade 2 *Regiella*. We found no evidence of differential gene expression in an aphid genotype from the *Trifolium* biotype: zero genes were differentially expressed in response to harboring *Regiella* strain .313 (which was confirmed using qPCR on three immune genes). Together, these results are consistent with a scenario where Clade 2 *Regiella* have evolved increased within-host density that harms host survival, and aphids from the *Trifolium* biotype have adapted to prevent immune suppression to control symbiont numbers. Our findings might more broadly suggest, then, that hosts and beneficial heritable microbes can evolve antagonistically, which has been suggested by other studies in aphids [9,58] and other organisms [59].

The phenoloxidase enzyme is required for the activation of melanogenesis in invertebrates. Against multicellular parasites, melanin is deposited around a foreign object via immune cells

(hemocytes), and the melanin capsule prevents the growth and reproduction of parasites [60]. *PO* is also upregulated in response to microbial pathogens in many studies [61,62] and it is thought that because phenoloxidase is cytotoxic it helps immune cells kill phagocytosed microbes [63]. It is important to note that regulation of phenoloxidase also occurs at the post-translational level [60], and future work is needed to link changes in gene expression in the presence of symbionts to protein levels. Aphid immune cells express phenoloxidase [64], are known to have phagocytic properties [64,65], and have been shown using microscopy to contain facultative symbionts including *Regiella* [65]. Harboring *Regiella* (and *Hamiltonella* but not other species of symbionts) leads to a sharp decrease in the numbers of circulating immune cells (called granulocytes) [65]. One possibility is that *PO* knockdowns via RNAi in our study disrupted the cellular immune responses aphids use to regulate symbionts during development, but the natural mechanisms *Regiella* might be using to suppress *PO* and other immune genes are unknown.

In addition to the effects on symbionts we have uncovered in this study, phenoloxidase has been shown to be an important part of the pea aphid's response to pathogens (e.g. fungal pathogens [66,67]). Functionally, a recent study found that silencing of *PO1* and *PO2* via RNAi leads to decreased resistance of pea aphids against pathogenic bacteria and a generalist fungal pathogen [68]. Together, these results show that the same molecular mechanisms are influencing interactions with both beneficial and pathogenic microbes in this system. The protection against specialist fungal pathogens conferred by *Regiella* might benefit hosts, but changes in gene expression in the presence of some strains of *Regiella* could trade-off with an increased risk of infection with other pathogens. Recent work on animal immune systems has emphasized the role of immune mechanisms in regulating mutualistic interactions between microbes and hosts. In general, how immune systems evolve to manage the complex task of interacting with distinct microbes with different effects on host fitness is an important question.

Finally, our findings emphasize the importance of host genetic variation in associations with beneficial microbes [69]. We found that hybrids between aphid biotypes harbor symbionts at intermediate densities to their parental lines and only partially differentially express immune genes. The extent to which a host responds to symbiont infection is therefore likely be a quantitative trait, much like resistance against pathogens, that is influenced by variation at multiple to-be-determined loci, and subject to natural selection on the relative costs and benefits of symbiosis. This work thus contributes to a growing view of animal microbiomes as complex phenotypes that influence animal fitness, which are to some extent under host control.

## Materials and methods

### Pea aphids and symbiont establishment

Pea aphids reproduce parthenogenetically under certain light and temperature conditions (16L:8D at 20˚C), allowing us to rear large numbers of genetically identical and developmentally synchronized individuals for use in experiments. Wild-collected lines were cured of their original symbiont infections using antibiotics [37] and maintained asexually in the lab for several years before use in experiments (S5 and S6 Tables specify collection information for aphid genotypes and symbiont strains).

Throughout these experiments we used established protocols to infect aphids with facultative symbionts [70–72]. We inject a small volume of hemolymph from an infected donor aphid into a 1st instar recipient using a glass capillary needle. We then rear these aphids to adulthood and then collect an offspring from late in the birth-order to establish the infected line. When this aphid produces offspring, we extract DNA (using a lysis buffer with proteinase K and an ethanol precipitation [73]) and screen the line for symbionts using PCR with

symbiont-specific primers [43] (S7 Table): (94˚C 2 min, 11 cycles of 94˚C 20s, 56˚C (declining 1˚C each cycle) 50 s, 72˚C 30 s, 25 cycles of 94˚C 2 min, 45˚C 50 s, 72˚C 2 min and a final extension of 72˚C 5 min). Each biological replicate in these experiments (an "aphid line") originated from a separate symbiont-injection and screening except where noted.

### Measurements of symbiont density using qPCR

We established aphid lines from the LSR1 genotype with two strains of *Regiella*: Clade 1 *Regiella* (strain .LSR), and Clade 2 *Regiella* (strain .313). These two strains of *Regiella* each come from one of the two main clades of *Regiella* found among pea aphids as determined using an established protocol for MLST sequence typing [45,74]. We reared lines that had successfully acquired a *Regiella* infection under asexual conditions for four generations, at which point we re-screened lines for *Regiella* infection. We then used qPCR to compare symbiont density between these strains. We removed embryos from groups of 7 adult aphids, and extracted DNA using the Qiagen DNEasy kit under recommended conditions. We used qPCR primers that amplify a conserved region of the *Regiella hrpA* gene (S7 Table). Amplification of *g3PDH* was used as an endogenous reference gene that controlled for the relative abundance of host DNA in each sample. Primer concentrations were optimized against a serial dilution of gDNA (400/350nM F/R and 300nM for *g3PDH* and *hrpA*, respectively). We calculated -$\Delta C_T$ values by $-(C_{T\ hrpA}-C_{T\ g3PDH})$ and analyzed these values with a t-test. Note that this approach reveals the relative density of symbionts relative to host tissue across different samples, but does not measure the absolute abundance of symbionts. Experimental data from this and all of the experiments described below can be found in S1 Experimental Data.

### Effects of *Regiella* on host gene expression using RNAseq

We then measured the effects of harboring *Regiella* on gene expression using RNAseq. We used the lines established above with either *Regiella* strain .LSR or .313, and symbiont-free aphids of the same host genotype. For the 'no symbiont' treatment, we sham injected aphids (injected aphids with a small volume of hemolymph (0.25μl) from an uninfected adult donor aphid) and handled aphids in the same way as with symbiont-injected aphids.

For transcriptome sequencing, we collected adult, fourth generation aphids on the first day that each line produced offspring and dissected and removed developing embryos (in order to measure gene expression of the mother without including RNA from her embryos). We stored carcasses in TRIzol (Invitrogen) at -80˚C. Each sample contained ~14 adult carcasses collected from multiple host plants. We extracted RNA using TRIzol-chloroform and an isopropanol precipitation with an ethanol wash. We digested genomic DNA and cleaned the RNA using the Zymo Clean & Concentrate–5 kit with the DNAse I enzyme. RNA quality control was conducted on a bioanalyzer chip, and 12 sequencing libraries (4 biological replicates x 3 treatments) were constructed using the NEBNext Ultra II RNA Library Prep Kit for Illumina (including poly-A selection and 15 rounds of PCR amplification). Libraries were sequenced across one lane of Illumina PE150 sequencing (approximately 20 million reads per library) with a 250-300bp insert per library.

### RNAseq analysis

We estimated the average insert size of paired-end libraries using Picard Tools v.2.21.3 in java 1.8.0, and mapped reads to the pea aphid reference genome v.2.1 [44] using tophat v.2.1.1 [75]. We counted reads mapped to each annotated gene (using a modified version of pea aphid genome annotation v.2.1 (https://bipaa.genouest.org/sp/acyrthosiphon_pisum/) with several duplicated genes removed from the file) using the count function in htseq v.0.9.1 [76] and the

'union' overlap mode (S8 Table). We analyzed read counts using EdgeR v.3.22.3 in R v.3.5.0. Genes with a minimum threshold of aligned reads, determined by the filterByExpr function in edgeR, were retained in the analysis. We fit a quasi-likelihood model to the data using the glmQLFit function, and we tested for statistically significant differential expression of each gene using a quasi-likelihood F-test, interpreting genes with a false discovery rate (FDR) of $< 0.05$ as differentially expressed in response to *Regiella* infection.

## Immune gene expression across facultative symbiont species via qPCR

We used qPCR to verify our RNAseq results, and to explore how differences in gene expression due to *Regiella* in key innate immunity genes varied across facultative symbiont species. We used qPCR primers (S7 Table) that amplified 80-120bp of two target genes of interest (*PO1*: ACYPI004484 and hemocytin: ACYPI003478, which were also found to be differentially expressed in the *Lotus* genotype, described below). We used four endogenous control genes (Glyceraldehyde 3-phosphate dehydrogenase (*g3PDH*): ACYPI009769, *NADH dehydrogenase*: ACYPI009382, *β-tubulin*: ACYPI001007, and *Rpl32*: ACYPI000074). Primer concentrations were optimized against a 1:10 serial dilution of gDNA (200ng– 0.2ng gDNA per reaction) to an efficiency of 100 +/- 10% (*PO1*: 100nM; hemocytin: 100nM; *g3PDH*: 400/350nM F/R; *NADH*: 350/300nM F/R; *β-tubulin*: 400nM; and *rpl32*: 200nM). Reactions were run on a Bio-RAD CFX96 Real-Time System machine, with an initial step of 95˚C for 3 minutes and 40 cycles of 95˚C for 10s and 60˚C for 30s. Each 20μL reaction included a 1X PCR buffer, $Mg^{+2}$ at 2mM, dNTPs at 0.2mM, EvaGreen at 1X, 0.025 units/μL of Invitrogen taq, and 40ng of cDNA. Three technical replicates were run for each reaction.

We measured expression of these genes in lines with and without symbionts in two separate experiments. First, we collected aphids from the *Regiella*-infected lines used for the RNAseq above (no symbionts, Clade 1 .LSR *Regiella*, and Clade 2 .313 *Regiella*). We dissected out and removed embryos, pooled adult carcasses, extracted and cleaned RNA, and DNAse treated samples as above. We synthesized cDNA using the BioRad iScript cDNA synthesis kit under recommended conditions. For each sample we averaged the $C_T$ values from the endogenous control genes, and calculated $-\Delta C_T$ values by $-(C_{T\ target} - C_{T\ mean\ endogenous\ control})$. We analyzed differences in gene expression between symbiont-free, Clade 1, and Clade 2 lines with one-way ANOVAs on the $-\Delta C_T$ values, and used Tukey HSD tests for pair-wise comparisons among different symbiont backgrounds. We performed separate analyses for the two genes.

In a second experiment, we injected three additional symbiont species into aphids and measured *PO1* and hemocytin expression as above. For donor aphids, we used an aphid line harboring *Serratia symbiotica*, a line harboring *Spiroplasma* sp. (strain .161), and two strains of *Hamiltonella defensa* (S6 Table). We only successfully established *Serratia* from two injection events after multiple attempts, and so the biological replicates of this assay were generated by splitting the lines onto multiple plants after 2 generations before sampling at generation 5; the other lines represent independently injected lines. We maintained sham-injected (symbiont-free) aphids under identical conditions as above. Gene expression was measured and analyzed as above. We note that the *Spiroplasma sp.* strain used in this experiment has been shown, like *Regiella*, to be protective against fungal pathogens while the other symbiont species used have not found to confer fungal protection. The two *Hamiltonella* strains used were collected in the same field and may not represent distinct symbiont genotypes from each other. Data were analyzed as above.

## Expression knock-down via RNAi

We designed primers that amplify regions of two target genes (531bp of *PO1* and 483bp of hemocytin) with the T7 promoter sequence (TAATACGACTCACTATAGGG) on the 5' end

of each primer using the e-RNAi Webservice (https://www.dkfz.de/signaling/e-rnai3/). Primer sequences can be found in S7 Table. We PCR amplified these regions from cDNA under recommended conditions. PCR products were sequenced using Sanger sequencing primed with the T7 promoter sequence to confirm target identity. We then purified 160μL of PCR product (using NaOAc and EtOH precipitation) and concentrated it to 500ng/μL. We used the MEGA-Script RNAi kit to synthesize dsRNA from PCR amplicons under recommended conditions and a 15hr transcription incubation. We ran dsRNA (at a 1/400 dilution) on a 2% agarose gel to verify that a single band was obtained of the correct size, and we then concentrated the dsRNA product to approximately 3300ng/μL using LiCl and an ethanol precipitation, and eluted the final dsRNA product in MEGAScript buffer. We repeated these protocols to generate dsRNA from lacZ as a control as in [77].

In a first experiment, we injected 1$^{st}$ instar (1-day-old) aphids with approximately 100μg dsRNA from either *PO1* or *lacZ* as a control using a glass capillary needle attached to a syringe on the underside of thorax of each aphid. We used two aphid lines that harbored either *Regiella* strain .LSR or .313. We collected injected aphids at two time-points after injection: at 72hrs and at day 9 when they had undergone their final molt to the adult stage. For the 72hr samples, groups of 3 whole aphids were pooled in TRIzol and stored at -80˚C for RNA extraction, or were stored in tubes at -20˚C for gDNA extraction. For the adult samples, we dissected out developing embryos and stored adult carcasses individually for RNA or DNA extraction, and we also pooled embryos from three adults and stored them in tubes for DNA extraction.

From the samples stored in TRIzol, we extracted RNA, synthesized cDNA, and measured the expression of *PO1* and four endogenous control genes using qPCR as above. We calculated -ΔC$_T$ values as above and analyzed these using two-way ANOVAs (with treatment (*lacZ* vs. *PO1*) and *Regiella* strain (.LSR vs. .313) as factors). We conducted post-hoc analyses using Tukey's HSD tests. The two time-points were analyzed separately. We extracted DNA from the remaining samples using the Qiagen DNEasy kit and measured *Regiella* densities using qPCR amplification of the *hrpA* gene as above. We analyzed these data using two-way ANO-VAs and Tukey's HSD tests as above in R v.3.5.1 after testing for model assumptions.

For adult injections, we reared LSR1 aphids with *Regiella* strain .LSR as above. We then injected adult aphids (9 days old) with 0.3μL of dsRNA (approximately 1μg total). In this experiment we performed knock-downs of two genes: *PO1* and hemocytin. We first collected aphids harboring strain .LSR at 72 hours after injection to validate our knock-downs. For each gene, we pooled groups of three aphids into 8 samples, removed developing embryos, extracted RNA from adult carcasses, synthesized cDNA, and measured gene expression as above. We analyzed -ΔC$_T$ values for each gene using t-tests with treatment as the independent variable. In this experiment we also measured expression of *PO2* in aphids that had been injected with dsRNA from *PO1* to verify that our knock-down was specific to *PO1*. We then injected adult aphids with approximately 1μg dsRNA, removed and discarded developing embryos after 72hrs, pooled samples into groups of 4 dissected aphids, extracted DNA as above, and measured *Regiella* densities using qPCR as above. We analyzed -ΔC$_T$ values from this experiment using a one-way ANOVA, and included treatment (*lacZ* control, *PO1*, or hemocytin) as a factor in the analysis. We conducted post-hoc analyses using a Tukey's HSD test to compare levels within treatment.

## RNAseq on aphid genotypes from multiple host-plant associated biotypes

We selected three aphid genotypes each from a different host-plant associated biotype (*Lotus corniculatus*, *Ononis spinosa*, and *Trifolium* spp.); information on collection location can be found in S5 Table. We established *Regiella* strain .313 infections in each line as above. Each

*Regiella*-infected line was established from a different symbiont injection and maintained separately on *V. faba* plants. In parallel, control aphids were sham injected as above. After 4 generations, we froze seven adult aphids from each line in liquid nitrogen and stored at -80˚C. RNA was extracted and purified as above. For the *Trifolium* biotype, RNAseq libraries and sequencing was conducted as described for the LSR genotype used above. For the *Lotus corniculatus* and *Ononis spinosa* biotypes, dual-indexed stranded sequencing libraries were constructed using the NEBNext polyA selection and Ultra Directional RNA library preparation kits. Libraries were sequenced on one lane of Illumina HiSeq 4000 (Paired-end 150bp) generating a target of > 2x 280M reads. We analyzed each genotype separately, comparing libraries with and without a *Regiella* infection, as above.

## F1 crosses

We used two of these biotype lines (C317 from *Trifolium* and 663 from *Lotus*) for the F1 genetic cross because we found from RNAseq data that these lines responded differently to *Regiella* infection. To induce male and female aphids for genetic crosses, we transferred stocks to 'autumn' conditions (short day, 13L:11D at 18˚C). After 30 days we moved third and fourth instar nymphs onto leaf-plates (a fava bean leaf in 2% agar in Petri dishes) to isolate virgin egg-laying sexual females (oviparae) and males. Oviparae have a characteristic thicker hind tibia, and this feature was used to isolate probable oviparae from males. The male screening was less stringent because virgin males were not needed. We setup each cross by placing the corresponding genotype oviparae and males onto a fava bean seedling, replenishing breeding stocks as they became available. After 24 hours, we treated melanized eggs with 10% calcium propionate to clean off the surface and then transferred eggs using fine-tipped forceps to a small petri dish with Whatman filter paper moistened with sterile water. We sealed the plates with parafilm wrap and left them in autumn conditions for a further 24 hours, after which dishes were transferred to a 2˚C incubator to diapause. After 3 months, eggs were removed from the diapause conditions and with fine-tip forceps, rolled against a Kimwipe to reduce any microbial growth. We then transferred diapaused eggs to a new leaf-plate and placed them in 'autumn' conditions (as above) until a fundatrix hatched. Each fundatrix was separated and a line was considered stable after two generations. We used this protocol to generate five F1 lines: three with line 663 as the maternal genotype and two with C317 as the maternal genotype.

## qPCR measures of gene expression in the F1 panel

We established replicate *Regiella* strain .313 infections in parental and the five F1 hybrid lines as above. After 4 generations, we collected three adults from each biological replicate, removed embryos via dissection, extracted DNA, and measured symbiont density all as above. We grouped the replicate F1 lines from each direction of the cross together to analyze these data, and used a one-way ANOVA in R version 3.5.0, comparing density among genotypes using a Tukey's HSD post-hoc test.

We then measured expression of four immune genes aphids with and without *Regiella* using qPCR in the parental and two of the F1 lines. Four generations after injection, four aphids from each biological replicate from each genotype were removed from plants, embryos were dissected out of adult carcasses and stored in TRIzol at -80˚C. We extracted RNA and synthesized cDNA as above. For each sample, we measured expression against the four endogenous control genes used above. Here we measured expression of four target genes: two copies of phenoloxidase (*PO1*: ACYPI04484 and *PO2*: ACYPI072244), hemocytin (ACYPI003478), and Nitric Oxide Synthase (ACYPI001689). $-\Delta C_T$ values were analyzed using an ANOVA after testing for model assumptions; post-hoc tests using the multcomp package in R v.3.5.0 were

conducted to compare the specific effect of *Regiella* on each genotype, and to compare interaction terms between *Regiella* presence/absence and host genotype. Analysis of expression of each gene was conducted separately.

## Supporting information

**S1 Table. Significantly differentially expressed genes from the RNAseq experiments.** Numbers show $\log_2$ fold changes of expression in aphids with vs. without *Regiella*. Host genotype and symbiont strain are indicated in the header.
(DOCX)

**S2 Table. Results of post-hoc tests (Tukey's HSD) analyzing gene expression of PO1 and Hemocytin (corresponding with Fig 1D and 1E).** The top section labeled "*Regiella*" shows the results of statistical analyses of lines harboring *Regiella*. The bottom section labeled "Other Symbiont Species" shows the results of statistical analyses of lines harboring one of the *Hamiltonella* strains, *Spiroplasma*, or *Serratia*. Statistical significance at $p < 0.05$, $p < 0.01$, and $p < 0.001$ is indicated with a *, **, or *** respectively.
(DOCX)

**S3 Table. Results of post-hoc tests (Tukey's HSD) analyzing *Regiella* densities in F1 lines.** Statistical significance at $p < 0.05$, $p < 0.01$, and $p < 0.001$ is indicated with a *, **, or *** respectively.
(DOCX)

**S4 Table. Results of post-hoc tests (Tukey's HSD) analyzing gene expression in F1 lines.** Significance at $p < 0.05$, $p < 0.01$, and $p < 0.001$ is indicated with a *, **, or *** respectively.
(DOCX)

**S5 Table. Collection information for the aphid genotypes used in this study.** *Reg* refers to *Regiella insecticola*; *Ham* refers to *Hamiltonella defensa*.
(DOCX)

**S6 Table. Collection information for the symbiont strains used in this study.**
(DOCX)

**S7 Table. Primer Sequences.**
(DOCX)

**S8 Table. Sequencing and alignment results.**
(DOCX)

**S1 Experimental Data.** Contains the data for each experiment as indicated in the tab headers.
(XLSX)

## Acknowledgments

Thanks to Will Brewer and Jennifer Keister for technical assistance, and Ailsa McLean and Jan Hrček for providing aphid lines used in this study.

## Author Contributions

**Conceptualization:** Benjamin J. Parker.

**Formal analysis:** Benjamin J. Parker.

**Funding acquisition:** Benjamin J. Parker.

**Investigation:** Holly L. Nichols, Elliott B. Goldstein, Omid Saleh Ziabari, Benjamin J. Parker.

**Writing – original draft:** Benjamin J. Parker.

**Writing – review & editing:** Holly L. Nichols, Elliott B. Goldstein, Omid Saleh Ziabari, Benjamin J. Parker.

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
