## [Decision Letter · Decision Letter 0]

11 Mar 2021

Dear Dr. Parker,

Thank you very much for submitting your manuscript "Intraspecific variation in immune gene expression and heritable symbiont density" for consideration at PLOS Pathogens. As with all papers reviewed by the journal, your manuscript was reviewed by members of the editorial board and by several independent reviewers. In light of the reviews (below this email), we would like to invite the resubmission of a significantly-revised version that takes into account the reviewers' comments.

The reviewers universally felt this was an interesting and convincing paper but raised a number of points to deal with in a revision. I would particularly highlight the question of whether you have demonstrated immune suppression, or if this an interpretation that should be made more cautiously in the discussion. It would seem to me that you have not demonstrated this, as this would require you to show active sabotaging of host immunity by the symbiont. This might require more nuanced discussion and wording.

We cannot make any decision about publication until we have seen the revised manuscript and your response to the reviewers' comments. Your revised manuscript is also likely to be sent to reviewers for further evaluation.

Sincerely,

Francis Michael Jiggins

Associate Editor

PLOS Pathogens

Renée Tsolis

Section Editor

PLOS Pathogens

Kasturi Haldar

Editor-in-Chief

PLOS Pathogens

orcid.org/0000-0001-5065-158X

Michael Malim

Editor-in-Chief

PLOS Pathogens

orcid.org/0000-0002-7699-2064

The reviewers universally felt this was an interesting and convincing paper but raised a number of points to deal with in a revision. I would particularly highlight the question of whether you have demonstrated immune suppression, or if this an interpretation that should be made more cautiously in the discussion. It would seem to me that you have not demonstrated this, as this would require you to show active sabotaging of host immunity by the symbiont. This might require more nuanced discussion and wording.

Reviewer's Responses to Questions

**Part I - Summary**

Reviewer #1: This manuscript focuses upon beneficial bacterial symbionts and their relationships with their aphid hosts, particularly from an immune modulation perspective. The bacterial symbionts provide benefits to the aphid hosts, but there may be competing interests on the part of the symbionts (for increased density and potential transmission) vs. the aphids (control of symbionts potentially harmful at high densities). These competing interests could be modulated by manipulation of the aphid immune system by either the aphids or the bacteria. This manuscript seeks to determine the genetic basis of any immune interactions by measuring gene expression differences in aphids with different strains of Regiella symbiont infections and also among different aphid genotypes. A small number of immune genes were found to change in expression, depending upon the symbiont type. The authors then knocked down expression of a few key immune genes to determine the impact upon symbiont density and found that one affects the other. Finally, the authors crossed some aphid genotypes together to determine how variation in genotype affected symbiont density and also immune gene expression. Their results showed that genotype influences both factors.

I found this manuscript very well put together. The feedback between beneficial symbionts and insect immune systems is a very interesting topic, and this work contributes new information on the impact of aphid immune gene expression upon symbiont density in hosts, showing that the aphid immune system is an important part of the relationship between insects and their symbionts.

Reviewer #2: In many host-symbiont relationships, there is variation across a population or populations in the titer of symbionts. However, the mechanisms that regulate these titers are not fully understood, particularly considering beneficial vs harmful symbionts. Here, Nichols et al. demonstrate that a strain of Regiella symbionts in aphids can increase its density through suppression of host immunity. This trait varies across host genetic backgrounds, potentially indicating that there is an arms race of adaptation and counter-adaptation between aphid and symbiont across populations and strains. This is a novel study in that it is one of the few to identify a mechanism of heritable symbiont regulation, and the mechanism is itself novel. The experiments are well executed with a strong design, the figures are visually appealing, and it is clearly written.

Reviewer #3: Nichols et al., present we well written report on the interaction between secondary symbionts of pea aphids and the host immune system. Specifically, using a thorough set of experiments, the authors clearly demonstrate the effect of Regiella secondary symbiont on the expression and function of several specific aphid immune genes and resultant symbiont titer. The strength of this work derives from the set of experiments that address their basic question through the cross-infection of aphid lines with different strains of Regiella to identify host immune genes that underlie the control of symbiont titer, experimental infection of aphid biotypes that have different host genotypes with Regiella strains to show specific host-symbiont genotype interactions, and the generation of hybrid lines to demonstrate that the trends are genetically controlled and heritable. Moreover, the authors further use RNAseq, RNAi, and qPCR validation to show the specific actions of immune genes in modulating symbiont titer. The authors propose that some Regiella strains are able to antagonistically increase their titer in some host genotypes through the suppression of certain host immune response genes (Phenoloxidase and Hemocytin). The authors interpret this as antagonistic co-evolution between a “beneficial” symbiont and the host

Overall, I found the presented work to be well done, well written, and very compelling. The manuscript is easy to read and clearly outlines the purpose, results, and implications of the work done. The figures are clear and demonstrate the results very well. I have only a few relatively minor comments that may be considered (through discussion) prior to publication:

**Part II – Major Issues: Key Experiments Required for Acceptance**

Reviewer #1: Insect phenoloxidase proteins are actually produced as pro-phenoloxidase enzymes that become activated to phenoloxidase by enzymatic cleavage. This means that their regulation primarily occurs at the post-transcriptional level by activation of a serine protease cascade induced by immune challenge. Therefore it is unclear what impact changes in the expression of the phenoloxidase genes will have upon aphid immunity. It would be very worthwhile to see how levels of phenoloxidase proteins change with different facultative symbiont infections and how much of the protein is in the active vs. inactive forms.

Was it possible to replicate experimental results? qPCR results are shown for a relatively small sample size, making it unclear if the same results would be obtained in a replicated experiment. It was not mentioned in the methods whether experiments were replicated and had qualitatively similar results, but it would be good to add this is possible.

I don’t think there’s enough evidence presented in this work to determine whether the differences in Regiella density observed between aphid lines is due to aphids manipulating their own immune responses or Regiella actively suppressing immune gene expression.

As I was reading the results, I highlighted the word “suppressed” on line 285, and was surprised by the choice of word here rather than something more unbiased.

When I reached the discussion I saw that the authors are making an argument that Regiella is responsible for manipulating host gene expression. Arguments related to high Regiella infection densities being maladaptive for aphid fitness (found in other studies) were used to support this conclusion. However, I wonder if the impact of Regiella upon fitness is also variable and correlated with frequencies of co-occurrence. How often does the LSR1 strain of aphids encounter Regiella from “clade 2”? Overall I found the arguments in the discussion somewhat confusing with respect to the data. The authors may be able to make their arguments clearer or perhaps treat both possibilities more equally.

Reviewer #2: There are no suggestions for major revisions.

Reviewer #3: (No Response)

**Part III – Minor Issues: Editorial and Data Presentation Modifications**

Reviewer #1: I find the indications of statistical significance in figures confusing, eg. Figure 2A and B, Figure 4C. It is clear that there are drastic differences between .LSR and .313 strains for phenoloxidase expression 8 days after infection. Is the line and the asterisk at the top of the figure meant to convey this, or that there are differences between lacZ and PO1 treatments? I believe that all statistically significant differences in the data should be indicated, even if it isn’t the main result that is being conveyed (within reason).

In figure 1D and E, I think the grey bars showing means are missing (they are in all other comparable figures)

References to figures are often incorrect in the text and should be corrected.

Figure 1 and Figure 3 are confusing regarding statistical cut-offs used. It would be nice to see how the FDR cut-offs correspond to the pvalues shown on the y axes. It seems logically unideal to use a FDR <0.05 cut-off for all comparisons except those shown in Figure 3A.

How are the LSR1, Lotus, Ononsis, and Trifolium lines related (closely or distantly)?

Reviewer #2: There are several minor questions and suggestions that would help enhance the paper if included in the text:

-Line 129: It would be beneficial to give a little more background information on the 313 strain here in the main text including host background, host plant, and collection location as an easy comparison to the lsr strain

-Figure 1D: Both of the Hamitonella strains also reduce phenoloxidase 1 at levels comparable to 313; are they also known to have relatively high densities?

-Line 165: It would help to briefly mention the rationale for using these two genes in particular as opposed to the other differentially expressed genes for rnai and qpcr experiments

-Line 170: In this section, it may help to mention/compare the tissue localization of these microbes for naïve readers (all the same or are there differences?)

-All figures: It would help to name the statistical tests and * values for all figures within the captions

-Figure 3: How closely related are the different aphid host backgrounds tested in this study- is there a pattern in relation to the Regiella densities? Which Regiella clades do these other host genotypes naturally harbor- are they all clade 2?

-Line 286: Italicize/capitalize Lotus

-Figure 4: What is meant by the letters at the bottom of the panels by the X axes/genotypes? I could not find the information in the caption. Why only two groups for trifolium x lotus in B vs three groups for the inverse?

-Figure 4B: Italicize Regiella on Y axis

-Figure 4C: Perhaps I missed this, but were statistics run to compare the expression differences within each genotype? Right now the stats are across genotypes, which is a helpful comparison, and the individuals with or without Regiella are differentiated visually. But, is there statistically a difference in gene expression within each genotype between Regiella vs no symbiont, and how do these differences compare across the lines? Visually, I can appreciate the trend, but the statistics would be helpful as well.

-It is interesting that the .313 Regiella and Trifolium type host were collected in the same place and the Trifolium type had clade 2 symbionts already, and that the Lotus type was collected with no symbiont. I found this information in the supplement, but I think it would be beneficial to directly mention this correlation in the discussion to bolster the adaptation-counteradaptation argument.

Reviewer #3: 1) Since the authors infer an antagonistic coevolution model, it might be worthwhile to add a bit more specific detail to the Introduction framing the antagonism between Regiella and aphid hosts. The fungal protection and general theory of host-symbiont antagonism is addressed here, but some introductory material setting the specific basis for the theory/hypothesis that drives this work would help the reader understand the complicated Regiella-aphid interaction better.

2) The authors reasonably logic that certain Regiella strains are the causative agent of suppressed immune response in some aphids. But, is there any information that can be gleaned from strain specific genomes to indicate, or permit speculation of, the specific bacterial mechanisms?

Alternatively, the authors collected a significant amount of RNAseq data from these systems. Could that data be mined to identify highly expressed and/or differentially expressed bacterial genes across their infection studies to hypothesize what mechanism are at play.

3) In figure 1D, it looks as though both Regiella strains causes decline in PO1. The biological significance of this is not well explained, nor discussed. I see that the effect of .313 is apparently stronger, but the other strain also seems to have a strong effect here. Is the ability of Regiella .313 to establish higher titers due to the bulk suppression of both PO1 and Hemocytin? Perhaps some clarifying points in the discussion can help put this together more strongly.

It also appears that other secondary symbionts have the same effect in suppressing PO1, but not-necessarily Hemocytin. Does this provide any insight on the combinatorial effect of Regiella .313 on suppressing both genes? Is it also known whether Hamiltonella and Regiella increase their titer through modulation of PO1? Some discussion here could be brought into the general antagonistic co-evolution theory, as well.

4) For the RNAi experiments illustrated Figs 2A-B, it seems that the lacZ control has a similar effect, or at least trend, as the PO1 dsRNA target and resultant Regiella titer… particularly over the long-term. This result is not really discussed. Does simply injecting the aphids, or the buffer used, cause a response in PO1 expression system with downstream effects on Regiella titer? Were any other controls used in this work, or in previous work, to verify the effect of RNAi delivery on the immune system response?

5) Finally, the antagonistic co-evolution hypothesis is not revisited in the discussion. It might be nice to summarize the results presented here in the broad theory as it pertains to this body of theory, insect immune system evolution, and host-symbiont interactions.

PLOS authors have the option to publish the peer review history of their article (what does this mean?). If published, this will include your full peer review and any attached files.

Reviewer #1: No

Reviewer #2: No

Reviewer #3: No
---

## [Editor Report · Decision Letter 1]

9 Apr 2021

Dear Dr. Parker,

We are pleased to inform you that your manuscript 'Intraspecific variation in immune gene expression and heritable symbiont density' has been provisionally accepted for publication in PLOS Pathogens.

Best regards,

Francis Michael Jiggins

Associate Editor

PLOS Pathogens

Renée Tsolis

Section Editor

PLOS Pathogens

Kasturi Haldar

Editor-in-Chief

PLOS Pathogens

orcid.org/0000-0001-5065-158X

Michael Malim

Editor-in-Chief

PLOS Pathogens

orcid.org/0000-0002-7699-2064

The revision includes a thorough response to all the reviewer's comments and numerous revisions. This is a valuable addition to the literature on insect symbionts.
---

## [Editor Report · Acceptance letter]

21 Apr 2021

Dear Dr. Parker,

We are delighted to inform you that your manuscript, "Intraspecific variation in immune gene expression and heritable symbiont density," has been formally accepted for publication in PLOS Pathogens.

Best regards,

Kasturi Haldar

Editor-in-Chief

PLOS Pathogens

orcid.org/0000-0001-5065-158X

Michael Malim

Editor-in-Chief

PLOS Pathogens

orcid.org/0000-0002-7699-2064